

# Upscaling surface energy fluxes over the North Slope of Alaska using airborne eddy-covariance measurements and environmental response functions

Andrei Serafimovich[1], Stefan Metzger[2,3], Jörg Hartmann[4], Katrin Kohnert[1], Donatella Zona[5,6], and Torsten Sachs[1]

[1]Helmholtz Centre Potsdam, GFZ German Research Centre for Geosciences, Telegrafenberg, D-14473 Potsdam, Germany
[2]National Ecological Observatory Network, Fundamental Instrument Unit, 1685 38th Street, Boulder, CO 80301, USA
[3]University of Colorado, Institute for Arctic and Alpine Research, 1560 30th Street, Boulder, CO 80303, USA
[4]Alfred Wegener Institute Helmholtz Centre for Polar and Marine Research (AWI), Am Handelshafen 12, D-27570 Bremerhaven, Germany
[5]Department of Animal and Plant Sciences, University of Sheffield, Western Bank, Sheffield S10 2TN, UK
[6]Department of Biology, San Diego State University, 5500 Campanile Drive San Diego, CA 92182, USA

*Correspondence to:* Andrei Serafimovich (andrei.serafimovich@gfz-potsdam.de)

**Abstract.** The objective of this study was to upscale airborne flux measurements of sensible heat and latent heat and to develop high resolution flux maps. In order to support the evaluation of coupled atmospheric/land-surface models we investigated spatial patterns of energy fluxes in relation to land-surface properties.

We used airborne eddy-covariance measurements acquired by the POLAR 5 research aircraft in June-July 2012 to analyze
surface fluxes. Footprint-weighted surface properties were then related to 21,529 sensible heat flux observations and 25,608 latent heat flux observations using both remote sensing and modelled data. A boosted regression tree technique was used to estimate environmental response functions between spatially and temporally resolved flux observations and corresponding biophysical and meteorological drivers. In order to improve the spatial coverage and spatial representativeness of energy fluxes we used relationships extracted across heterogeneous Arctic landscapes to infer high-resolution surface energy flux maps, thus
directly upscaling the observational data. These maps of projected sensible heat and latent heat fluxes were used to assess energy partitioning in northern ecosystems and to determine the dominant energy exchange processes in permafrost areas. This allowed us to estimate energy fluxes for specific types of land cover, taking into account meteorological conditions. Airborne and modelled fluxes were then compared with measurements from an eddy-covariance tower near Atqasuk.

Our results are an important contribution for the advanced, scale-dependent quantification of surface energy fluxes and
provide new insights into the processes affecting these fluxes for the main vegetation types in high-latitude permafrost areas.

## 1   Introduction

Arctic ecosystems are undergoing very rapid changes as a result of warming climate (Chapin et al., 2005; Serreze and Barry, 2011) and their response to climatic change has important implications, not only on local to regional scales (McFadden et al., 1998; Chapin et al., 2000) but also on a global scale (Bonan et al., 1992; Foley et al., 1994). Thawing permafrost has the





potential to release large quantities of carbon dioxide and methane that are currently trapped in frozen soil. Microbes may also produce increasing amounts of methane as more organic material becomes available due to thawing. The Arctic is likely to be affected by changes to the timing of snow-melt, to the length of the growing season, to the vegetation, and to precipitation regimes. The regional energy budget of Arctic ecosystems can be changed, both directly or indirectly, through a lower albedo as

a result of reduced snow cover (Euskirchen et al., 2007, 2010), or a higher albedo due to the changes in vegetation (Randerson et al., 2006). Liu and Randerson (2008) provided evidence that fire-induced changes in the surface energy budget also contribute to regional cooling at high latitudes through an increase in surface albedo during spring and summer. The sensible heat flux (H) and latent heat flux ($\lambda$E), which together form a major part of the surface energy budget, therefore have a marked effect on climatic variability and associated feedbacks.

Surface energy partitioning is an important physical process that has a strong influence on the ground heat flux and hence on the thermal condition of Arctic ecosystems. Direct measurements of surface fluxes are usually made using eddy-covariance (EC) flux towers (Baldocchi et al., 2001). Energy fluxes have been previously investigated in different polar regions using a variety of techniques. Vourlitis and Oechel (1999) analyzed surface fluxes and the energy budget of a tussock tundra ecosystem in Alaska; they reported a strong correlation between daily fluctuations in evapotranspiration and daily fluctuations in net

radiation, as well as a predominance of biological limitations to evapotranspiration over meteorological limitations, during the measurement period. Westermann et al. (2009) and Langer et al. (2011a, b) used independent measurements of radiation and heat flux and documented the annual cycle of the surface energy budget on on Svalbard and Samoylov Island in the Lena River Delta; both of these sites are high-arctic permafrost sites. The relative importance of different budget components over a full year was also investigated. The ratio of H to $\lambda$E, which is known as the Bowen ratio, was found to vary between 0.25 and 2,

depending on the water content of the uppermost soil layer (Westermann et al., 2009). Beringer et al. (2005) investigated surface energy fluxes measured at Council, on the Seward Peninsula of Alaska, at five sites representing the major vegetation types in the transition zone from Arctic tundra to forest, these being tundra, low shrub, tall shrub, woodland (treeline), and boreal forest sites. Changes in vegetation structure that increased sensible heat flux were shown to enhance warming in northern high latitudes. Ueyama et al. (2014) evaluated changes in regional surface energy fluxes due to fire and spring warming in Alaska

between 2000 and 2011, based on an upscaling of EC tower measurements, and highlighted the importance of these processes in amplifying or reducing Arctic warming over decadal time scales.

EC tower measurements may, however, only be representative of small areas immediately surrounding the tower locations (Kaharabata et al., 1997; Schuepp et al., 1992). Moreover, due to the lack of infrastructure EC towers are scarce and unevenly distributed over high latitude permafrost wetlands, which makes it difficult to use EC tower measurements for accurate model

upscaling from regional to global flux contributions from the Arctic. Airborne measurements can be used as an alternative way to investigate surface exchange at regional scales (Desjardins et al., 1995). Metzger et al. (2013) used airborne flux measurements and developed a procedure to estimate the sensible heat and latent heat fluxes for different land covers in a heterogeneous landscape. This method extracts environmental response functions (ERFs), which establish a relationship between spatially or temporally resolved flux observations and environmental drivers. Dobosy et al. (2017) analyzed airborne

data in the space and time domains using the flux fragment method (FFM) and compared the theory behind the FFM with that



behind the wavelet method. An improved random-error estimate was proposed that takes into account the serial correlation of the time/space series and the heterogeneity of the signal. Sayres et al. (2017) used the FFM method to analyze regional-scale drivers of the heterogeneity and variability of methane fluxes measured by a small, low-flying aircraft over the North Slope of Alaska.

Since changes in climate-related parameters such as evaporation, precipitation, and land cover, can have a significant effect on the regional surface energy budget, a good understanding of how energy fluxes in the Arctic will respond to climatic changes is crucial. In this study we aimed to upscale airborne flux measurements and to develop spatially extensive, high resolution flux maps that could be used to provide new insights into surface exchange processes and to validate coupled atmospheric/land-surface models. Particular emphasis was placed on a detailed analysis of airborne EC measurements and the spatial patterns

of surface energy exchange across the North Slope of Alaska. In this paper we attempt to answer the following particular questions: (i) Which surface properties are the main drivers for energy fluxes in permafrost areas? (ii) Is it possible to use relationships extracted across heterogeneous Arctic landscapes to create high-resolution surface flux maps and to directly upscale observational data with minimal assumptions? (iii) How large are land-cover-specific energy fluxes under particular meteorological conditions and what are the energy partitioning patterns in northern ecosystems? Lastly, airborne and modelled

fluxes are compared with EC tower measurements and the factors leading to discrepancies are discussed.

The rest of this paper is organized as follows. The study area and climate are first described (Sect. 2.1). The experimental set-up and the state-of-the-art processing of airborne EC measurements are then presented in Sect. 2.2. Sect. 2.3 provides a summary of the model configuration and model data used for the flux upscaling. Section 2.4 explains how a nonparametric machine learning technique was used to upscale direct flux measurements across the North Slope of Alaska. The potential of

the extracted relationships between flux observations and surface properties are evaluated in Sect. 3. The ERFs of the energy surface fluxes are first presented in Sect. 3.1. The variability of energy fluxes between different northern ecosystems and energy partitioning within northern ecosystems are discussed in Sect. 3.2 and 3.3. The airborne flux measurements are compared with the modelled fluxes in Sect. 3.4. The final section (Sect. 4) presents our conclusions and discusses possible improvements and applications of the presented methods.

## 25  2   Material and methods

### 2.1   Study area

The following analysis focuses on the North Slope of Alaska, a large terrestrial area at latitudes greater than 69°N, bordered to the north by the Arctic Ocean (the Chukchi Sea to the north-west and the Beaufort Sea to the north-east) and to the south by the Brooks Range. The investigated area covers 87,160 km$^2$, extending 330 km in an east-west direction and 275 km north-south;

it consists mainly of coastal plains to the north and foothills to the south, which differ in their climate and topography as well as in their vegetation (both structure and composition).

According to Zhang et al. (1996), the North Slope of Alaska can be divided into three main climate zones which they referred to as the Arctic foothills, Arctic inland, and Arctic coastal zones. The climate is strongly influenced by both continental and





marine environments. Cloud cover, fog, and north-easterly winds are common over the coastal zone between June and August, while the inland area experiences higher average air temperatures, more variable wind directions, and more frequent clear sky conditions.

The mean monthly temperatures over the North Slope of Alaska are below 10°C. Only between June and August are average
air temperatures above the freezing point and the annual mean temperature is below -10°C. Precipitation in the coastal zone is of the order of 150 mm, increasing towards the south, and the tundra is covered with snow for about 9 months of the year. The mean annual wind speed is about 6 ms⁻¹. The active above the permafrost layer is about 300 to 400 mm thick (Wendler et al., 2010). The predominant forms of vascular vegetation on the North Slope are tundra shrubs and graminoids (Walker, 2000).

## 2.2   Airborne eddy-covariance measurements

An airborne survey to measure methane fluxes was carried out across the North Slope of Alaska from 28 June to 2 July, 2012 (AIRMETH-2012: airborne measurement of methane fluxes), based out of Barrow, Alaska (71°18′N, 156°46′W). The research aircraft, a POLAR 5 (Hartmann et al., 2018) belonging to the Alfred Wegener Institute (AWI) Helmholtz Centre for Polar and Marine Sciences, flew at low altitudes measuring fluxes along horizontal transects totaling more than 3,115 line kilometers (about 41 flight hours) over the North Slope of Alaska. Forty vertical profiles were also obtained to estimate the height of the
planetary boundary layer. The results presented in the following analysis are representative for the period from 10:00 hours local time (LT = UTC-8 h) to 14:00 hours LT, which we refer to as the "reference perio". Flight lines are shown in Fig. 1 and and the four time intervals used in our analysis are summarized in Table 1. These time intervals are characterized by air temperatures between 5 and 11°C and a light breeze blowing from the north-west, or from the north-east, east, or south-east.

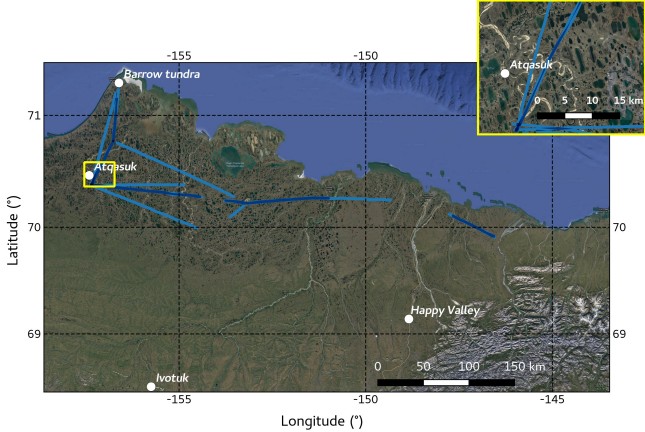

**Figure 1.** Flight lines from the 2012 airborne survey over the North Slope of Alaska that were used in the analysis. The dark blue flight lines were more frequently surveyed than the light blue lines. The insert shows the location of the EC tower in Atqasuk that was used for the comparison in Sect. 3.4. Map data: Google, DigitalGlobe.





**Table 1.** Details of the POLAR 5 survey flights carried out in 2012 over the North Slope of Alaska, the time intervals used in the analysis, and median values for meteorological parameters averaged over these time intervals.

| Flight date | Start time (LT) | End time (LT) | Time used for the analysis (LT) | Median in-situ temperature (°C) | Median horizontal wind speed ($ms^{-1}$) | Median wind direction (°) |
|---|---|---|---|---|---|---|
| 28.06.2012 | 13:43 | 18:02 | 13:43 - 14:05 | 5 | 2.1 | 306 |
| 29.06.2012 | 09:22 | 16:39 | 09:52 - 13:54 | 6 | 4.8 | 79 |
| 30.06.2012 | 10:59 | 14:19 | 10:59 - 14:03 | 9 | 2.6 | 150 |
| 02.07.2012 | 13:21 | 16:58 | 13:21 - 13:43 | 11 | 4.6 | 53 |

The POLAR 5 aircraft was equipped with a nose boom carrying a Rosemount 5-hole probe to measure different wind components. A PT100 sensor was installed in an unheated Rosemount housing at the tip of the nose boom to measure the air temperature. A HMT-330 sensor (Vaisala, Helsinki, Finland) to measure the humidity of the air was also mounted in a Rosemount housing. Data were recorded at 100 Hz. A CR2 chilled mirror hygrometer (Buck Research Instruments LLC,

Aurora, Colorado, USA) providing highly accurate (but slow) absolute values was used to validate humidity measurements. The aircraft movements and altitude were acquired by a Laseref V Inertial Navigation System (Honeywell International Inc., Morristown, New Jersey, USA), with the position derived using a Global Positioning System (NovAtel Inc., Calgary, Alberta, USA). The aircraft was also equipped with a KRA 405B radar altimeter (Honeywell International Inc., Morristown, New Jersey, USA), an LD90/RIEGL laser altimeter (Laser Measurements Systems GmbH, Horn, Austria), and a CMP22 pyranometer (Kipp

& Zonen B.V., Delft, the Netherlands). The median altitude for the survey flights was 38 m above ground level and the median true airspeed was 69 $ms^{-1}$.

To estimate the energy fluxes between the earth's surface and the atmosphere we followed Metzger et al. (2013) and used a modified version of their time-frequency-resolved eddy-covariance method in an early version of the edd4R eddy-covariance data processing software (Metzger et al., 2017). The spikes were first removed from the raw turbulence data and the sampling

frequency reduced from the 100 Hz of the original data to a 20 Hz resolution, using block averaging. Computations were made using a continuous wavelet transform to enable a 100 m spatial discretization of the flux measurements. This was achieved by integrating the wavelet cross-scalograms in frequency over transport scales up to 20 km, and in space using a 1,000 m moving window along the flight paths, in 100 m steps. This allowed the calculation of spatially resolved turbulence statistics and of sensible heat and latent heat fluxes for overlapping subintervals of 1,000 m length, with a 100 m resolution. The flux

data was subjected to quality assurance and quality control measures, which included a steady state test (Foken and Wichura, 1996; Vickers and Mahrt, 1997) to detect non steady state conditions during the selected perturbation time scale, and an ITC (Integral Turbulence Characteristics) test (Foken, 2008a) to compare the measured integral turbulence characteristics with the modelled characteristics. Data with quality flags from 1 to 6 were retained for subsequent analysis. The subintervals were centred above each cell of the remote sensing data overflown by the POLAR 5 aircraft. Footprint-weighted surface properties,

which preserve the continuous nature of the information content, were subsequently determined for a total of 21,529 sensible



heat flux observations and 25,608 latent heat flux observations. The footprint model used was the Metzger et al. (2012) 2-D version of the Kljun et al. (2004) 1-D model.

## 2.3 Configuration and evaluation of the WRF model

The Weather Research and Forecasting (WRF) model was used to simulate the potential temperature, the dry mole fraction of water vapor, the shortwave down-welling radiation, and the height of the planetary boundary layer. These atmospheric drivers were used to project the earth's surface to atmosphere exchange of sensible heat and latent heat throughout the North Slope of Alaska. The WRF model is a numerical weather prediction model designed for use on a regional scale (Skamarock et al., 2008), that can be used for operational forecasting and atmospheric research. It is, however, adaptable to a higher resolution (1 km or less) by using a nested domains technique and zooming in to the area of interest. For our analysis we used the WRF-ARW (Advanced Research WRF core) version 3.2.1; the configuration of the WRF model is given in Table 2. The WRF model was initialized using two nested domains, D1 and D2, with spatial resolutions of 3 km and 1 km, and temporal resolutions of 3 h and 30 min, respectively (Fig. 2). The meteorological input data were obtained from the final global gridded analysis archive of the National Center for Atmospheric Research (1999), which had a $1° \times 1°$ spatial resolution and six hours temporal resolution. Sea surface temperatures with a $0.5°$ spatial resolution were provided by the National Centers for Environmental Prediction (NCEP; Gemmill et al., 2007).

**Table 2.** Configuration of the WRF model domains and physical parametrizations

|  | Domains and physical parameterizations |
| --- | --- |
| dx, dy [m] | 3000 (D1); 1000 (D2) |
| Microphysics | Lin (Purdue) scheme (Lin et al., 1983) |
| Longwave radiation | Rapid Radiative Transfer Model (Mlawer et al., 1997) |
| Shortwave radiation | Goddard shortwave scheme (Chou and Suarez, 1994) |
| Surface layer | MM5 similarity theory surface layer scheme (Paulson, 1970; Dyer and Hicks, 1970; Webb, 1970; Beljaars, 1994) |
| Land surface | Noah Land Surface Model (Chen and Dudhia, 2001) |
| Planetary boundary layer | Yonsei University scheme (Hong et al., 2006) |
| Cumulus parameterization | Kain-Fritsch scheme (Kain, 2004) |

Figure 3 shows weather conditions during the reference period. The synoptic situation was characterized by air temperatures close to zero over the Arctic Ocean, rising to $\approx 20°$ in the southern part of the study area. Close to the coast the wind blew mainly from the north-east, swinging round to blow from the south or south-east close to the Arctic foothills; north-westerly winds were observed over the Barrow area on 28 June. The wind speed was between 1 and 4 ms$^{-1}$ indicating light breezes.





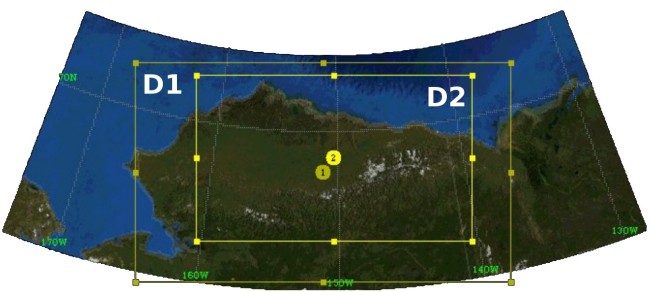

**Figure 2.** Location of the D1 and D2 nested domains.

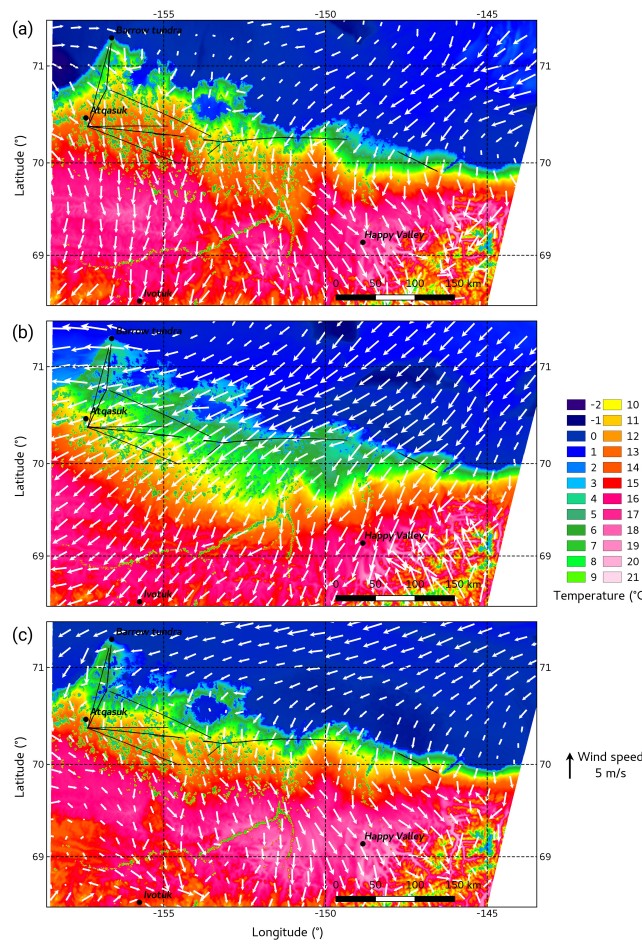

**Figure 3.** Air temperature at 2 m above the ground and wind speed at 10 m above the ground, simulated by the WRF model for 28 June at 14:00 hours LT (a), for 29 June at 12:00 hours LT (b), and for 30 June at 12:00 hours LT (c). Black lines represent POLAR 5 flight lines.





## 2.4 Estimation of environmental response functions

A boosted regression tree (BRT) technique (Metzger et al., 2013; Elith et al., 2008) was used to estimate environmental response functions (ERFs) between spatially and temporally resolved flux observations and the corresponding biophysical and meteorological drivers. The BRT technique is a non-parametric machine learning technique that attempts to learn a response by

observing inputs and their associated responses, finding dominant patterns (regression trees), establishing a response function according to the coherencies in the training data, and then adaptively combining large numbers of relatively simple tree models to optimize the predictive performance. An example of the BRT method is shown in Fig. 4.

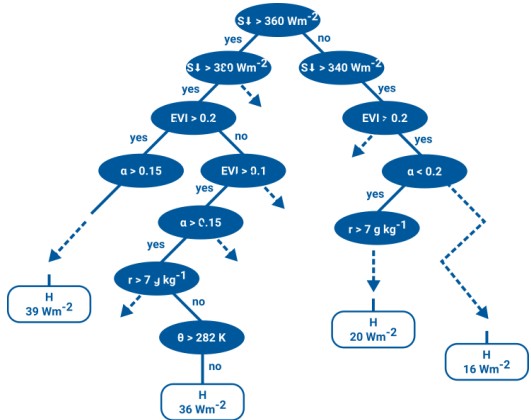

**Figure 4.** Example of boosted regression trees (BRT) learning a response of sensible heat flux (H) to observations of the downward shortwave solar radiation (S↓), the enhanced vegetation index (EVI), the mixing ratio (r), and the land surface albedo ($\alpha$).

To train the model we used remote sensing data, meteorological state variables from WRF modelling, and airborne measurements. The remote sensing data came from the Moderate Resolution Imaging Spectroradiometer (MODIS), post-processed by

the National Research Council (NRC) of Canada (Trishchenko et al., 2006; Luo et al., 2008). We used bi-linear interpolation to increase the spatial resolution to 100 m and linear interpolation in time to obtain a separate map for each flight day. The flux footprints were subsequently used to link surface properties with the corresponding measured energy fluxes. In order to take into account the altitude dependency of surface fluxes the ratio of the measurement height ($z_m$) to the height of the planetary boundary layer ($z_{ABL}$), estimated by the WRF model, was used as a training parameter. Using WRF data allowed us to mitigate

the assumption of horizontally homogeneous meteorological states (Metzger et al., 2013), which is clearly violated in our study area, as shown in Fig. 3. The temporal variations in the surface fluxes were taken into account by using the time of observation as a training parameter. The mid-point time for each flight line was used as the time for the projection. A full list of the drivers tested is provided in Table 3.





**Table 3.** Biophysical and meteorological drivers used for estimating environmental response functions, and the corresponding data sources.

|  | Data source | |
| Parameter | Response | Projection |
| --- | --- | --- |
| Enhanced vegetation index EVI | MODIS MOD13Q1 | MODIS MOD13Q1 |
| Land surface albedo $\alpha$ | NRC SW BB Albedo | NRC SW BB Albedo |
| Downward shortwave solar radiation S↓ | POLAR 5 | WRF |
| Potential temperature $\theta$ | POLAR 5 | WRF |
| Mixing ratio r | POLAR 5 | WRF |
| Daytime | Observation time | Projection time |
| Ratio of measurement height $z_m$ to the height | POLAR 5, WRF | 5% of $z_{ABL}$ |
| of the planetary boundary layer $z_{ABL}$ | | |

## 3 Results and discussion

### 3.1 Environmental response functions of energy fluxes

BRTs can provide deep insights into ecologically complex interactions. These can be visualized using fitted ERFs that show the effect on surface fluxes of a specific state variable over its entire range, while all other state variables are held at their means. The ERFs for sensible heat flux are shown in Fig. 5 and for latent heat flux in Fig. 6. The most important factors affecting surface heat fluxes are S↓, EVI, and $\alpha$, all of which yield almost linear responces within the 10-90% range of the data distribution, and $\theta$ and r, which yield non-linear responces.

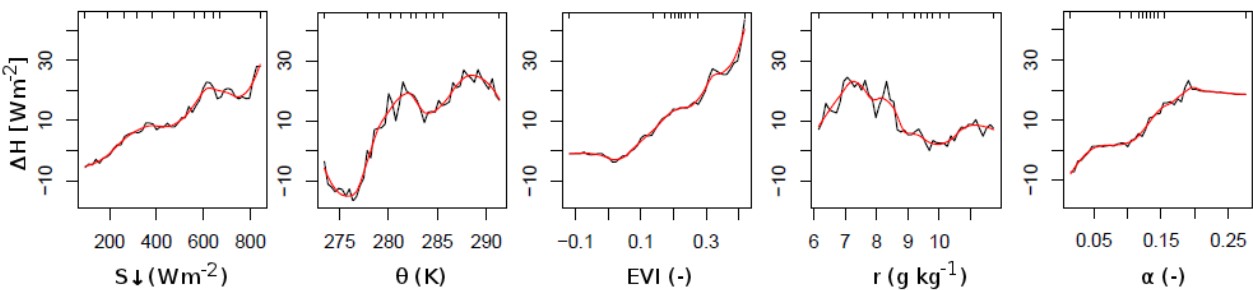

**Figure 5.** Environmental mean response functions for the sensible heat flux. The functions show the responses to changes in the downward shortwave solar radiation (S↓), potential temperature ($\theta$), enhanced vegetation index (EVI), mixing ratio (r), and land surface albedo ($\alpha$). The black line shows the variable response of the BRT and the red line is an equidistantly smoothed representation of the black line. Rug plots along the top margins of the plots show the distribution of the variables in deciles.





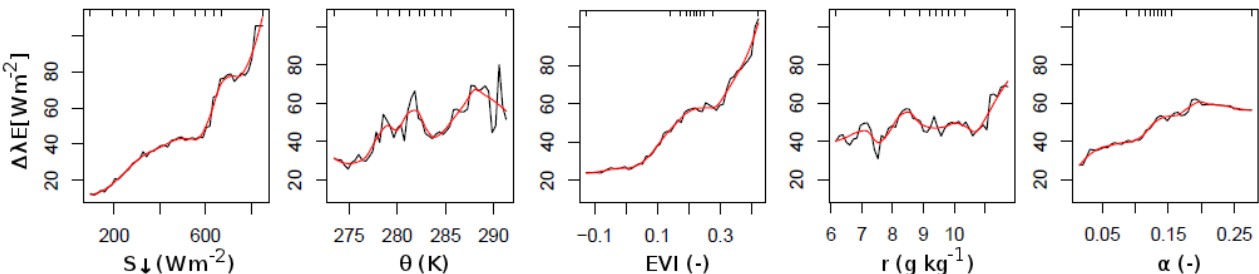

**Figure 6.** Environmental mean response functions for the latent heat flux. The functions show the responses to changes in the downward shortwave solar radiation (S↓), potential temperature ($\theta$), enhanced vegetation index (EVI), mixing ratio (r), and land surface albedo ($\alpha$). The black line shows the variable response of the BRT and the red line is an equidistantly smoothed representation of the black line. Rug plots along the top margins of the plots show the distribution of the variables in deciles.

Figure 7 shows a scatter-plot with hexagonal binning of the measured airborne values and BRT predicted values for sensible heat (a) and latent heat (b) fluxes. Both the observed H and $\lambda$E are in a good agreement with the BRT fitted values for fluxes up to 100 Wm$^{-2}$, with a slight underestimation by the BRT technique for values greater than 100 Wm$^{-2}$. The median absolute deviations in the residuals for the sensible heat and latent heat fluxes are less than 8% and 3%, respectively, and the coefficient

5 of determination (R$^2$) is greater than 0.99 in both cases. Metzger et al. (2013) showed that underestimations mostly occur along short sections of the flight lines that have highly intermittent solar irradiance. Finally, the resulting environmental response functions were used to extrapolate the sensible heat and water vapor exchange over spatiotemporally explicit grids of the Alaskan North Slope, using the remote sensing data and model output data as biophysical and meteorological drivers. In order to match the remote sensing data the WRF gridded data were down-scaled from the finest domain to a 100 m spatial resolution

10 using bivariate interpolation, and a bias adjustment was made of WRF atmospheric variables to match the in-situ airborne survey data.

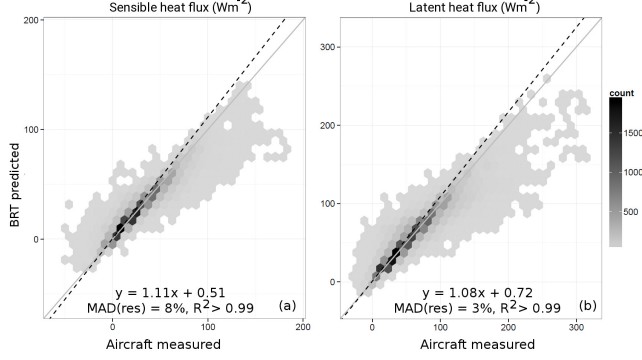

**Figure 7.** Scatter-plot with hexagonal binning of the measured (airborne survey) and BRT predicted sensible heat (a) and latent heat (b) fluxes. Altogether 21,529 data points were used for the sensible heat flux scatter plot and 25,608 for the latent heat flux.





### 3.2 Variability of energy fluxes between northern ecosystems

The BRT technique was used to extrapolate sensible heat and latent heat fluxes across the North Slope of Alaska. Separate flux maps for each flight line were created using a trained BRT model, together with meteorological data for corresponding times from the WRF model and remote sensing data. Median values were calculated from the individual maps and used to produce the ensemble maps in Fig. 8, which illustrate the spatial variability of energy fluxes across the North Slope of Alaska, well captured by ERFs. The latent heat flux varies considerably and shows a strong gradient from 160-180 Wm$^{-2}$ in the south to 10-20 Wm$^{-2}$ in the north, whereas the sensible heat flux has a less pronounced south-north gradient, with maximum values of 60-80 Wm$^{-2}$ in the south-western part of the study area and 10-60 Wm$^{-2}$ elsewhere. The airborne measurements obtained by Oechel et al. (1998) along the 148°55′W line of longitude between 68°55′N and 70°30′N, also indicated a decreasing trend in sensible heat and latent heat fluxes from south to north.

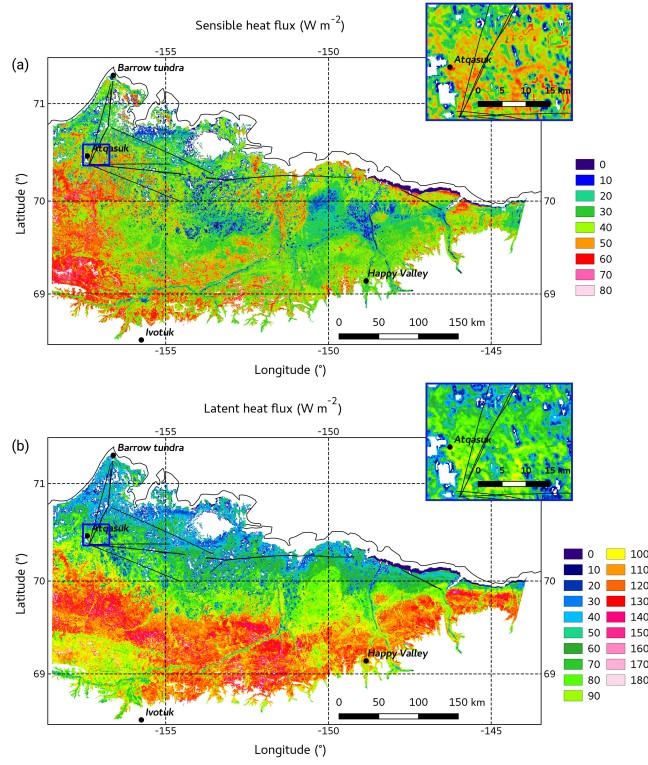

**Figure 8.** Median sensible heat (a) and latent heat (b) fluxes over the North Slope of Alaska, averaged over the reference period. Only those fluxes with a standard error < 30% are shown. The insert shows the location of the EC tower in Atqasuk that provided the measurements used for the comparison in Sect. 3.4. Black lines represent POLAR 5 flight lines.





The upscaled latent heat fluxes are comparable to those reported in previous publications. Latent heat fluxes measured by Oechel et al. (1998) were of the order of 100 Wm$^{-2}$ in the southern part of the survey area and close to 50 Wm$^{-2}$ in the northern part of the area. The averaged sensible heat flux measured by Oechel et al. (1998) was of the same order as the average latent heat flux, whereas the sensible heat fluxes derived in our study along the same path surveyed by Oechel et al. (1998) have

less variability and only range between 10 and 40 Wm$^{-2}$. This discrepancy may be due to the different times of day and dates of the measurements, to cloudiness, to variations in the EVI (as a proxy for soil moisture), and also to the different altitudes of the two aircraft during the flux measurements. The median altitude in the POLAR 5 survey was 38 m while the measurements obtained by Oechel et al. (1998) were from an altitude of 10-20 m. Possible reasons of flux inconsistencies will be discussed in Sect. 3.4.

Specific energy fluxes for different land cover classes (Table 4) were derived by combining high resolution surface flux maps (Fig. 8) with the National Land Cover Database (NLCD) data from 2011 (Homer et al., 2015) shown in Fig. 9. The averaged latent heat flux was 2-3 times greater than the averaged sensible heat fluxes for all land cover classes. A high latent heat flux of 112-113 Wm$^{-2}$ was found over vegetation types located in the southern part of the North Slope, such as dwarf shrubs (i.e. shrubs less than 20 centimeters high, with the shrub canopy typically comprising more than 20% of the total

vegetation) and shrubs/scrub (i.e. shrubs less than 5 m high, with the shrub canopy again typically comprising more than 20% of the total vegetation). Moderate fluxes (57-83 Wm$^{-2}$) were projected over herbaceous sedge (sedges and forbs, generally comprising maore than 80% of the total vegetation), barren areas (bedrock scarps, talus, glacial debris, strip mines, and gravel pits, where vegetation generally accounts for less than 15% of the total cover), and emergent herbaceous wetlands (where perennial herbaceous vegetation comprises more than 80% of the vegetative cover and the soil or substrate is continuously

saturated or covered with water). The lowest latent heat fluxes (30 Wm$^{-2}$ and 46 Wm$^{-2}$) were projected over open water (areas with less than 25% vegetation or soil cover) and perennial ice/snow (perennial cover of ice and/or snow, generally comprising more than 25% of the total cover), respectively. The relative proportions of each land cover class therefore need to be taken into account when considering flux uncertainty. Less representative land cover classes appear only rarely in flux footprints and were therefore less frequently used than the more representative classes for model training. The spatial pattern of the

projected latent heat flux (Fig. 8b) closely matches the spatial pattern of the land cover map and air temperature (Figs. 9 and 3, respectively), indicating a strong influence of these parameters over the latent heat flux. Sensible heat flux showed less variability over different land cover classes but was found to be highest over dwarf shrub vegetation, with moderate fluxes projected over herbaceous sedge, shrubs/scrub, emergent herbaceous wetlands, and barren land, and only low fluxes projected over open water and perennial ice/snow. The spatial pattern of the projected sensible heat flux (Fig. 8a) is more complicated

than that of the projected latent heat flux indicating that there are additional processes influencing the sensible heat flux.

Eugster et al. (2000) analyzed available results from long-term (one or more years) and short-term surveys and summarized the summer surface energy budget for a range of Arctic tundra and boreal ecosystems. Their mean fluxes for July were selected where the data time series were long enough and used for comparison. The lowest sensible heat and latent heat fluxes reported by Eugster et al. (2000) were measured over the large, deep Toolik Lake (10 Wm$^{-2}$ and 13 Wm$^{-2}$, respectively), whereas our

ERF projected energy fluxes for open water ecosystems were 13 Wm$^{-2}$ and 33 Wm$^{-2}$ higher, respectively, because they are



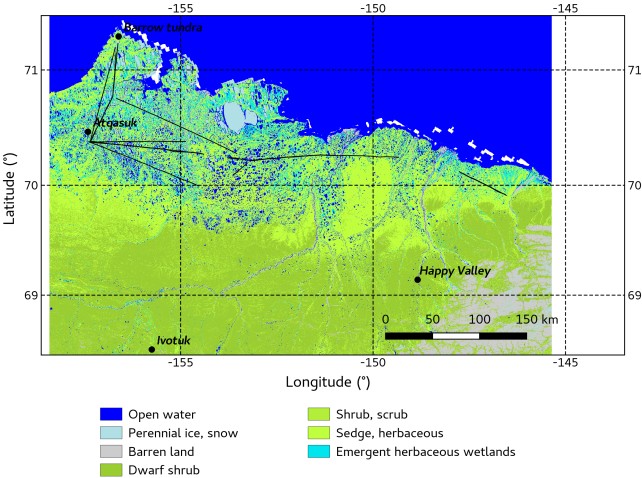

**Figure 9.** Land cover classes according to the National Land Cover Database, 2011. Black lines represent POLAR 5 flight lines.

**Table 4.** Relative coverage for each land cover class and median, maximum, 25% percentile, and 75% percentile of energy fluxes for different NLCD land cover classes, calculated from the ensemble flux maps shown in Fig. 8.

| Wetland class | Coverage | Sensible heat flux (Wm$^{-2}$) | | | | Latent heat flux (Wm$^{-2}$) | | | |
|---|---|---|---|---|---|---|---|---|---|
| | (%) | 25% | Median | 75% | Maximum | 25% | Median | 75% | Maximum |
| Emergent herbaceous wetlands | 9.4 | 25 | 34 | 42 | 107 | 41 | 57 | 76 | 207 |
| Herbaceous sedge | 42.5 | 28 | 37 | 45 | 111 | 60 | 83 | 101 | 216 |
| Shrub, scrub | 4.0 | 31 | 36 | 42 | 96 | 100 | 112 | 122 | 219 |
| Dwarf shrub | 34.5 | 35 | 41 | 51 | 117 | 101 | 113 | 122 | 221 |
| Barren land | 1.6 | 20 | 30 | 38 | 96 | 47 | 68 | 88 | 200 |
| Perennial ice, snow | 1.3 | 9 | 18 | 29 | 100 | 18 | 30 | 48 | 180 |
| Open water | 6.7 | 14 | 23 | 33 | 100 | 29 | 46 | 68 | 211 |

representative of different types of lakes, including small, shallow lakes. The sensible heat and latent heat fluxes measured by the EC tower over a sedge ecosystem near Happy Valley (22 Wm$^{-2}$ and 80 Wm$^{-2}$, respectively) differ from the ERF projected fluxes for the herbaceous sedge ecosystem by 15 Wm$^{-2}$ and 3 Wm$^{-2}$, respectively. The sensible heat and latent heat fluxes in Eugster et al. (2000) that were measured by multiple EC towers over shrub ecosystems ranged from 25 to 63 Wm$^{-2}$ and 33 to 93 Wm$^{-2}$, respectively. The ERF projected sensible heat fluxes lie within the same range but the ERF projected latent heat flux for shrub ecosystems is higher. This could be due to higher evapotranspiration rates as a result of the warm air temperatures observed during the reference period over the southern part of the North Slope of Alaska, where dwarf shrubs and scrub are more common.



### 3.3 Energy partitioning in northern ecosystems

The Bowen ratio ($\beta$) can be used as an indicator of an ecosystem's energy contributions to the regional climate. Figure 10 shows the spatial variability of $\beta$ derived from the median projected surface energy fluxes shown in Fig. 8. All data with latent heat flux values above the uncertainty of 10 Wm$^{-2}$ have been plotted. The maximum value of $\beta$ was found to be 4.03. Figure 10

indicates that evapotranspiration is the dominant process in the surface energy exchange over most of the area and $\beta$ varies from values close to zero up to one, i.e. this is a freely evaporating area. Only close to the coast does the sensible heat exchange predominate and $\beta$ exceed 1.3. Westermann et al. (2009) showed that variations in $\beta$ are closely related to the water content of the surface soil layer. In this area evapotranspiration from the coastal wetlands is restricted by cold surface temperatures and the amount of moisture available is limited by the thinness of the active layer overlying the permafrost (Eugster et al., 2000).

Similar observations have previously been reported by Harazono et al. (1998). Under the cold and humid meteorological situation influenced by the Arctic Ocean, the latitudinal temperature gradient over high-latitude ecosystems increases and leads to a high sensible heat exchange at the coast. Therefore $\beta$ increased by more than 1.5. In contrast, warm, dry atmospheric conditions increase evapotranspiration and $\beta$ therefore decreased to values of less than 1.

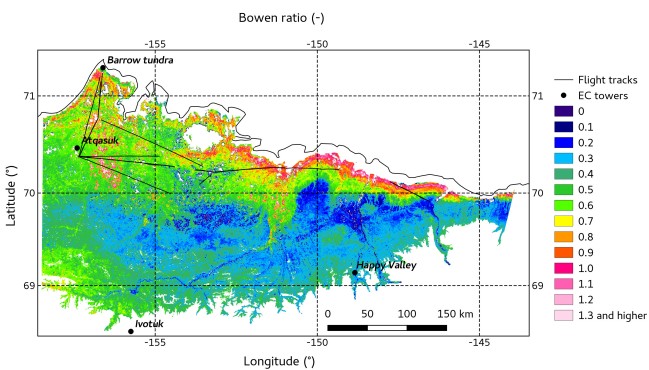

**Figure 10.** Median Bowen ratio ($\beta$) over the North Slope of Alaska, averaged over the reference period. Black lines represent POLAR 5 flight lines.

Superimposing the $\beta$ map (Fig. 10) on the NLCD 2011 land cover map (Fig. 9) allowed us to derive $\beta$ for the reference

period that were specific to particular types of land cover. The $\beta$ values were between 0.33 and 0.62 (see Table 5), which is within the range found in published literature. For example, Eugster et al. (2000) summarized typical ranges of $\beta$ for different Arctic ecosystems. $\beta$ of Arctic wetlands, low Arctic shrub tundra, and low Arctic coastal tundra were found to range from 0.2 to 0.7, 0.3 to 5, and 0.6 to 2.1, respectively. This is in in agreement with the $\beta$ values for emergent herbaceous wetlands and dwarf shrubs presented in this study. The spatial variations in $\beta$ in response to different meteorological conditions also lie

within these ranges. $\beta$ of emergent herbaceous wetlands, which are continuously saturated or covered with water, is close to the Bowen ratio for perennial ice/snow or open water. $\beta$ for areas of herbaceous sedge and dwarf shrubs, which can be periodically or seasonally wet and/or saturated, were found to be lower then the ratio for emergent herbaceous wetlands, but higher than





that for shrubs/scrub. The low $\beta$ values and small median deviations estimated for shrubs, dwarf shrubs, and scrub, which cover 38.5% of the investigated area, indicate that these ecosystems are important regulators of water loss to the atmosphere.

**Table 5.** Median Bowen ratio ($\beta$) values and median absolute deviation (MAD) of $\beta$ for different NLCD land cover classes, estimated from $\beta$ map (Fig. 10).

|  | Bowen ratio | |
| --- | --- | --- |
| Land cover class | Median | MAD |
| Emergent herbaceous wetlands | 0.58 | 0.15 |
| Herbaceous sedge | 0.48 | 0.22 |
| Shrub, scrub | 0.33 | 0.07 |
| Dwarf shrub | 0.37 | 0.10 |
| Barren land | 0.43 | 0.20 |
| Perennial ice, snow | 0.62 | 0.25 |
| Open water | 0.53 | 0.25 |
| Throughout the entire study area | 0.42 | 0.18 |

## 3.4 Comparison of surface energy fluxes derived from airborne survey, WRF modelling, and EC tower measurements

Realistic modelling of surface exchanges requires accurate representation of surface-atmosphere interactions, which means that the turbulent fluxes of energy and matter exchange must be accurate reproduced. Precise modelling of surface fluxes requires accurate simulation of the planetary boundary layer and fluxes need to be calculated using appropriate model parametrization. The modelled and measured meteorological parameters of the planetary boundary layer and turbulent energy fluxes were compared in order to test the performance of the WRF model. Data from the EC tower at Atqasuk (70°28′10.6′′N, 157°24′32.2′′W), 100 km south of Barrow, were available for the period of the airborne survey (Goodrich et al., 2016). Surface fluxes derived from the WRF model were compared with those derived from the POLAR 5 airborne survey and from the EC tower measurements. The modelled data were averaged over 9 grid cells (300 m x 300 m) around the tower. The POLAR 5 aircraft traverses between about 4-7 km to the east of the tower and we averaged those fluxes that were measured not more than 7 km from the tower and had less than 10 min time difference.

The measuring site represents wetland complexes that consist primarily of fens, dominated by moist nontussock sedges, prostrate dwarf-shrubs, and mosses, which are usually present in the slightly elevated hummocks and rims of low-centered ice-wedge polygons (Walker et al., 2005). Measurements were made at a tower height of 2.25 m. Wind velocity and sonic temperature were also measured with using a Solent R3 sonic anemometer (Gill Instruments Ltd., Lymington, UK) sonic anemometer at a height of 2.28 m. An To measure water vapor LI-7200 gas analyzer (LI-COR Biogeosciences, Nebraska, US) was used for water vapor measurements.




Figure 11 shows the measured and modelled surface fluxes, together with boundary-layer meteorological parameters. As can be seen in Figure 11(a,b), on 28th June, 1st July and 2nd July 2012 the sky at Atqasuk was almost cloud free, short-wave radiation was up to 700 Wm$^{-2}$, and the maximum air temperatures were about 12 or 13°C. The synoptic situation on 29th and 30th June 2012 was cloudy with a maximum air temperature of about 11 or 12°C. The airborne radiation measurements are in

5   agreement with those from the tower. The relative humidity reached a maximum of 90-95% at night, dropping to 65-70% at around midday or later. These trends in temperature and relative humidity were also observed by the POLAR 5 aircraft but the WRF model overestimated the short-wave radiation on 29th and 30th June 2012 and the sensible heat flux is therefore highly overestimated by the model on these particular days (Fig. 11d). The sensible heat fluxes measured by POLAR 5 are lower (median absolute deviation 81 Wm$^{-2}$) and the latent heat fluxes slightly higher (median absolute deviation 26 Wm$^{-2}$) than those measured by the EC tower (Fig. 11d,e).

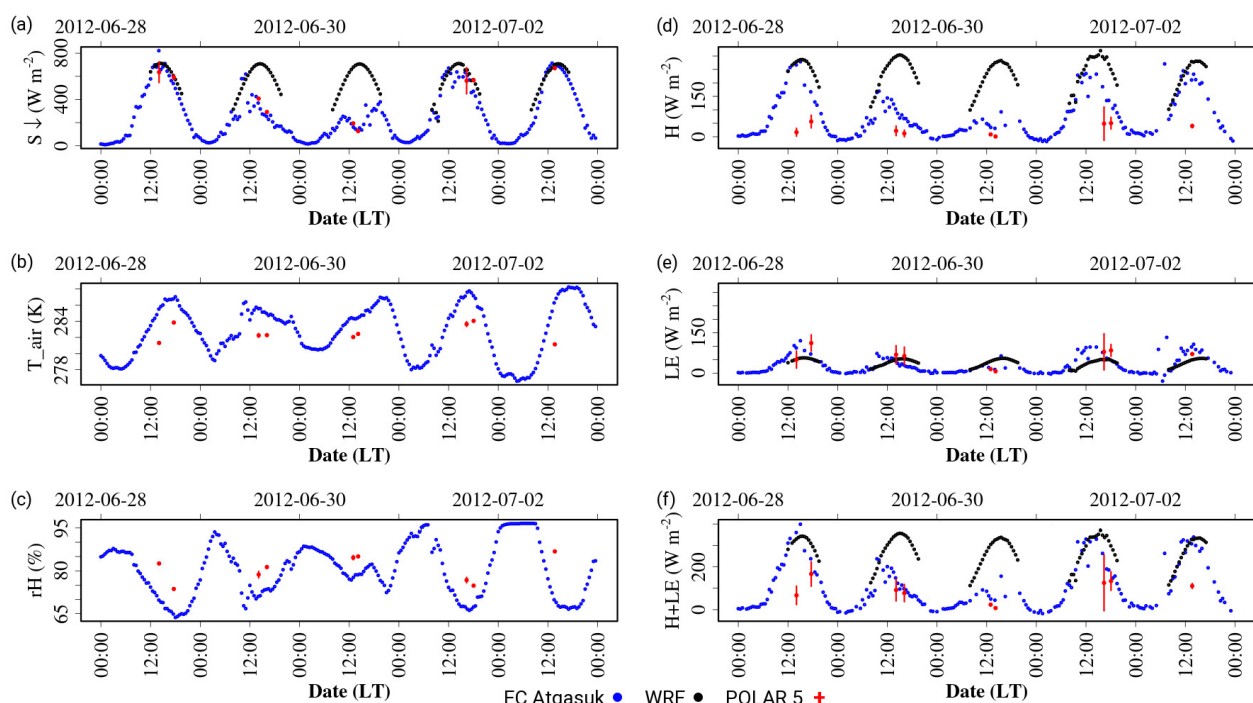

**Figure 11.** Left frame: short-wave radiation (a), air temperature (b), and relative humidity (c). Right frame: sensible heat flux (d), latent heat flux (e), and sum of the sensible heat and latent heat fluxes (f) based on measurements from the EC tower at Atqasuk (blue), the POLAR 5 airborne survey (red), or output from the WRF model (black). Red error bars indicate mean absolute deviations of the averaged POLAR 5 data.

Many previous investigations have also reported lower airborne sensible heat fluxes and higher airborne latent heat fluxes than those derived from EC tower measurements (Desjardins et al., 1992, 1995, 1997; Oechel et al., 1998; Gioli et al., 2004). Oechel et al. (1998) showed that sensible heat flux derived from EC tower measurements was generally higher than that




measured by all airborne surveys, but that latent heat flux showed a temporally more variable trend, with the EC tower fluxes being higher during June surveys and slightly lower during August surveys.

A summary of possible reasons for the discrepancy between fluxes measured by airborne surveys and those derived from EC towers can be found in Mahrt (1997). The airborne and tower data are collected from different levels and the storage and

advection can lead to height dependency in turbulent fluxes. As described above, we addressed this discrepancy by introducing the ratio of aircraft measuring height to boundary-layer height as a parameter in the ERF projected maps.

As reported by Sun and Mahrt (1994), surface energy budgets measured from aircraft seem to be more accurate if mesoscale fluxes are included, because the scale of horizontal flux increases with altitude and significant flux may occur where turbulence occurs on a scale greater than 2 km. The wavelet decomposition used in our data processing yields a high spatial resolution

for the flux observations and takes into account significant flux contributions from large eddies (2-4 km across), which are "invisible" for tower-based systems due to insufficient sampling of large-scale atmospheric movements. Foken (2008b) showed that exchange processes on the larger scales of a heterogeneous landscape have a significant influence on the energy balance closure. By including these fluxes, the energy balance can be approximately closed.

The footprint of tower measurements is smaller than that of airborne flux measurements. Aircraft measure turbulent fluxes

over different surfaces from an EC tower due to land surface heterogeneity. The footprint of the POLAR 5 survey in the vicinity of the EC tower had a width of between 800 m and 3.6 km, and it therefore "sees" a more averaged flux that is representative of the landscape as a whole, whereas the tower only "sees" a relatively small area. Sensible heat flux figures derived from the EC tower measurements were noticeably higher than those from the POLAR 5 survey under conditions of high incoming radiation. This can be explained by the larger proportion of wet surfaces within the POLAR 5 footprint area and the fact that

dry surfaces heat up more rapidly and to a higher level than wet surfaces, resulting in increased sensible heat flux. This can also be confirmed by considering the sum of both energy fluxes (Fig. 11f), which tends to be in agreement with flux data derived from EC tower measurements when the incoming shortwave radiation is high.

During the AIRMETH 2012 survey some lakes were partly covered by ice; the surface water temperature was therefore close to zero and 12°C lower than the air temperature at the time of high sensible heat fluxes. Due to the stable layer over the

water surfaces, turbulent fluxes can be directed to the surface, whereas over dry surfaces they are directed upwards. This leads to low averaged airborne fluxes, but high locally measured turbulent fluxes. A similar compensation of fluxes on a regional scale and the discrepancy between those fluxes and fluxes derived from EC tower measurements were also noted during the SHEBA experiment, and reported by Overland et al. (2000).

## 4 Conclusions

Upscaling of regional-scale flux measurements into regional or continental flux inventories is a useful way to improve our understanding of regional and global climatic changes. In this study we used POLAR 5 airborne turbulence measurements to upscale sensible heat and latent heat EC fluxes over the North Slope of Alaska, using the machine learning and boosted



regression tree technique. We have shown that this method can be used to isolate and quantify significant surface properties and to extend airborne flux observations to a regional scale, thus producing high-resolution surface flux maps.

The downward shortwave solar radiation, potential temperature, enhanced vegetation index, mixing ratio, and land surface albedo were found to be the most important parameters driving energy exchange processes between the land surface and the

atmosphere in permafrost areas. The resulting environmental mean response functions indicate linear responses of surface heat fluxes to changes in the downward shortwave solar radiation, the enhanced vegetation index, and the land surface albedo, and non-linear responses to changes in the potential temperature and the mixing ratio. The comparison of measured fluxes with predicted fluxes indicated the potential for using environmental response functions to extend airborne flux measurements to a regional scale, and quantitatively linking flux observations in the atmospheric surface layer to meteorological and biophysical

drivers in the flux footprints reveals a good agreement with median absolute deviations in the residuals of less than 8% and 3% for the sensible heat and latent heat fluxes, respectively. The coefficient of determination ($R^2$) was greater than 0.99 in both cases.

To overcome the disadvantage of the method presented in Metzger et al. (2013), which used the median meteorological state variables during each flight pattern to upscale airborne flux measurements, we utilized the Weather Research and Forecast

model simulations of the driving meteorological parameters. This improved the ability of their method to capture the spatial variability of energy fluxes across the North Slope of Alaska. The maps of energy fluxes were projected with a high spatial resolution of 100 m x 100 m. Marked regional differences were detected showing the non-uniform distribution of surface fluxes. High resolution flux maps allow land-cover-specific energy fluxes to be estimated, which can be used to validate coupled atmospheric/landsurface models. Our results show a strong south-north gradient in the latent heat exchange if cold weather

conditions prevail in the north and warm conditions in the south, with winds blowing from the Arctic Ocean. Sensible heat exchange is lower and has a less pronounced south-north gradient.

Energy partitioning information and the Bowen ratio are critical components of micrometeorological, climatic, and hydrological models and are widely used for comparing the surface energy balances of different climate zones and vegetation types. Our investigations into energy partitioning in northern ecosystems confirmed that, under the meteorological conditions of the

measuring period, evapotranspiration was one of the main process in the surface energy exchange over almost the whole of the North Slope. Only close to the coast was the evapotranspiration restricted and sensible heat exchange prevalent. The low Bowen ratio values derived for shrub, dwarf shrub, and scrub ecosystems indicate that they are important regulators of moisture loss to the atmosphere. The higher evapotranspiration capacity associated with such ecosystems results in a predominance of latent heat exchange over sensible heat exchange.

The spatial representativeness of flux tower measurements was checked and these data compared with the modelled and airborne fluxes. The airborne sensible heat fluxes were found to be lower than those measured by the tower, and small differences were observed in the latent heat fluxes. These discrepancies can be explained by the different heights at which the data was collected, where storage and advection can lead to height dependency, and the fact that the footprint of airborne flux measurements is more representative for the landscape as a whole. However, more measurements are needed covering different

meteorological situations in order to improve the machine learning, verify our results, and validate the model data.



The results obtained provide a valuable contribution to the advanced, scale dependent quantification of surface energy fluxes over extensive areas of terrestrial permafrost and reveal the potential of the upscaling method. The presented data set is unique for heterogeneous Arctic landscapes due to the exclusive use of airborne data, which are more representative on a regional scale than EC tower measurements. High resolution flux maps for Arctic areas, such as those presented herein, are scarce: they can

be used to validate modeling results and improve our understanding of physical processes related to permafrost-atmosphere interactions in Arctic landscapes.

*Competing interests.* The authors declare that they have no conflict of interest.

*Acknowledgements.* This work has received funding from the Helmholtz Association of German Research Centres through a Helmholtz Young Investigators Group grant to T.S. (Grant VH-NG-821), and is a contribution to the European Union's Horizon 2020 research and

innovation programme under grant agreement No. 727890, as well as to the Helmholtz Climate Initiative (REKLIM - Regional Climate Change). The AIRMETH airborne survey was fully funded by the Alfred Wegener Institute. The National Ecological Observatory Network is a project sponsored by the National Science Foundation and managed under a cooperative agreement by Battelle Ecology, Inc. This material is based on work supported by the National Science Foundation (Grant DBI-0752017). Any findings, opinions, conclusions or recommendations expressed in this paper are those of the authors and do not necessarily reflect the views of the National Science Foundation. We thank Junhua

Li and Shusen Wang at the National Research Council Canada and Yi Luo at Environment Canada for providing their version of MODIS remote sensing data products (Trishchenko et al., 2006). The authors wish to gratefully acknowledge Sebastian Wieneke for his support in post-processing remote sensing data. We also acknowledge Ke Xu, University of Wisconsin, Madison, WI, U.S.A. for providing the plotting algorithm used to create Fig. 5 and 6.





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
