# Peer review of "Upscaling surface energy fluxes over the North Slope of Alaska using airborne eddy-covariance measurements and environmental response functions"

_Atmospheric Chemistry and Physics, 2017_

## Referee Comment (RC1) · Anonymous Referee #1 · 25 Apr 2018

This is a very promising synergy of airborne turbulence measurement, machine-learned environmental response functions, and mesoscale modeling (WRF) to project air-surface exchanges of water and heat out over large areas of the North Slope of Alaska. It looks to be a significant advance in addressing this important issue.

Assets

1. Airborne campaign was generally well designed including straight tracks with multiple passes over some tracks and frequent profile ascents to find the mixed-layer struc-

ture and depth. Instrumentation and sample rate (100/s at about 70 m/s true airspeed) are appropriate to the mission. One apparent deficiency was in radiation. Although the investigators measured shortwave insolation, they did not mention measuring reflected shortwave (they did determine the albedo), or upwelling and downwelling long-wave (infrared) radiation.

2. Pg 5 line 17: "Wavelet cross-scalograms were integrated in frequency over transport scales up to 20 km" This raises the question of the length of the fight tracks. From the map (Figure 1) they appear to be mostly about 100 km, which still provides an adequate 80 km of flux uncontaminated by edge effects at the largest width (20km?) of the cone of influence.

3. Section 2.3: It's good to see a mesoscale model used in the study. In principle it can describe the mesoscale environment of the airborne campaign. It is difficult to relate model results quantitatively to what the aircraft and any fixed surface sites are reporting, especially in Alaska where the input data are relatively sparse. But for a "projection" (see item 6) it appears to work well.

4. Page 6, line 6: The verb "project" is a great word to describe the inference of air-surface exchange over large areas from measurements over small areas because it implies some sort of model which represents knowledge and draws on data both of which grow more sparse as the scale increases. "Upscaling" is commonly used but is troublesome because it implies that the system is (largely) scale-independent, like going from a model to a full-scale prototype.

5. Page 8: MODIS has something like 500 m to 1000 m pixels, which appears a good match to the km-scale spatial averages of the fluxes derived from this campaign's measurements.

6. Figures 5 and 6, pages 9 and 10: The "rug" plots showing the distribution of the variables in deciles are appreciated.

Questions and Issues

1. Page 5 and thereabouts: Recognizing flight safety as the ultimate first priority, did they next prioritize near-constant height above ground or did they try to minimize pilot adjustments? These are often a trade-off because terrain is rarely really flat. Minimizing control pressures improves fidelity of the flux measurements by minimizing the flow distortion, but usually requires higher-altitude flight to clear the terrain safely. Higher altitude gives up the advantage of cleaner sampling of heterogeneous surface fluxes.

2. Page 5, line 25 into Page 6: I presume that a sample of sensible or latent heat flux is one of the 1000 m mean fluxes computed every 100 m, perhaps further segregated by land-surface type. Because of the deep overlap, these are strongly autocorrelated implying fewer degrees of freedom (DF) than there are fluxes in the sample. There are ways to estimate this reduction in DF by determining the decorrelation length. A coarse guess would be to divide the sample size by 10 since the reporting interval (100 m) is one tenth of the averaging length (1 km). The loss is probably not that much. At the least the loss of DF by autocorrelation should be mentioned qualitatively.

3. Page 8, Fig 4: in the second-row left node the dashed arrow intrudes into the ellipse partially obscuring the criterion for insolation making it hard to read (S > 380 W/m^2?).

4. Pg 11 Figure 8 Caption: Is it proper to call the quality parameter a standard error, or is it more an uncertainty? If an error, what constitutes "truth"?

5. Figure 7 is interesting. There are two slopes of BRT predicted vs aircraft measured. The majority of the aircraft data match the BRT predicted quite well, but the majority of the spread of BRT vs measured has a shallower slope. Away from the training data, the BRT underpredicts the strong fluxes and overpredicts the weak fluxes. Were measures taken to avoid overfitting?

6. Page 17, line 7ff: It is indeed likely that under free-convection conditions (light winds and strong insolation) that fixed sites do not experience the larger patterns that aircraft can readily penetrate. If these structures are in fact turbulence (hence "random") they can be treated by integral techniques such as time/frequency-domain methods or time/space-domain averages. If they are not random it is not clear that an individual aircraft track in space and time will capture the relevant (nonrandom) structure unless it can be designed to do so.

7. Page 19, lines 2 and 3: The data set is no longer unique. Sayres et al. (2017), cited in this manuscript, also used an aircraft to measure fluxes (primarily latent heat and methane) over heterogeneous Arctic landscapes. Exclusive use of airborne data can be considered a liability because of the inherently greater uncertainty in airborne measurements. Having a surface reference, as in this manuscript with Atqasuk and Happy Valley, is important both for temporal continuity and for sanity check. Data are, of course, the more readily acquired from aircraft than from fixed sites over remote areas difficult of access as on Alaska's North Slope.

Copy-editing items (a few were found and noted, not guaranteed to be exhaustive list)

1. Page 4 line 16: "reference period" 2. Page 14 line 15: Likely: "periods that were" 3. Page 14 line 20: Sentence starts with Greek letter (beta for Bowen Ratio). Recasting of the sentence is recommended. 4. Page 15 line 19 Remove "An"; insert "an": "measure water vapor <an> LI-7200 gas analyzer..."; remove "for water vapor measurements. 5. Page 17 line 25: Turbulent fluxes over water surfaces are more likely to be suppressed (due to lack of both mechanical and buoyant generation) than to be directed to the surface. 6. Page 17 line 30: "Project" (verb) would be a really great word here instead of "upscale."

---

## Referee Comment (RC2) · Anonymous Referee #2 · 17 May 2018

General report on Edited version

This is an excellent combination of aircraft flux measurements, machine learning algorithms extrapolated to surface response functions. It also links to large scale modeling to allow contextual interpretation with respect to hydrological and energy budget response focused on an important climate sensitive region. It provides a significant advance in the area of airborne flux measurements and relevance to validation of surface response function dependent models. This is an important paper as this very thorough approach to airborne eddy covariance fluxes has really been missing from the scientific literature over the past 1-2 decades in general. The regional model comparison is a nice addition to capture the mesoscale variability and scales although the study is limited by the surface data availability. I do like the terminology used for model-aircraft comparison.

Updates
Relevant discussion on the design and implementation of the aircraft campaign is included and is sufficient for replication and addressing of issues and potential artifacts in such approaches.

The methodologies are very well described and relevant to the technique applied. These are appropriate to the conclusions arrived at with some limitations however in completing the full energy budget. These could have been discussed further with respect to the uncertainties but generally I don't think this could be improved on.

The relevant transform scales based on the flight track described appear consistent with the approach and is explained well and are also consistent with results previously published in the literature (although these were limited in terms of surface site comparisons). The relevant edge effects associated with the wavelet analyses are always an issue but I think these are within the uncertainties when scaling to the regional observations and looks quite reasonable. Whilst data quality control is critical for such wavelet analyses and could be quantified further I think we can assume this is good based on the results. There are other transform approaches that could have been compared but likely these would not have changed the results.

Page 8:
The addition of the distribution "rug" plot, Figure 5, is very useful.

*Minor Questions and Formatting Issues*
Item 1. Some brief comment on the appropriate optimization of relevant straight and level sampling altitudes for the flux measurements (discussed page 5 etc) with respect to heterogeneity scales within the flight track would be helpful but not essential here?

Item 2. Figure 4. Some of the arrows in the boosted regression tree figure overlap/obscure the text in the various nodes, e.g. a > 0.5, $S\downarrow$ > 380 W m$^{-2}$, r> 7 k kg$^{-1}$,

Item 3. Figure 7. It appears obvious that there are two clusters within the sensible and latent heat flux (predicted versus aircraft measured) domain with significantly different slopes with under-predictions at high values in each case. Can the authors comment on this? Is there a potential bias here?

Item 4. Figure 8. Legend: Has the standard error used in this figure been defined?

Item 5. As mentioned, the impact of enhanced convective conditions suggests potential under-sampling bias of all relevant scales in these conditions. It would be useful to mention the range therefore where such comparisons may break down, but this may require more detailed spectral analysis for another discussion. However I think this caveat/statement addresses the issue adequately for the work presented.

Item 6. Figure 7 is a brave plot (and we need more of them in the literature before relying overly on tower data). I think the discussion and literature references regarding the discrepancies with WRF are adequate but do highlight that there is still a great deal of work to do here.

Final comment: The authors are to be commended for delivering an excellent set of results.

---

## Author Comment (AC1) · 14 Jun 2018

This is a very promising synergy of airborne turbulence measurement, machine-learned environmental response functions, and mesoscale modeling (WRF) to project air-surface exchanges of water and heat out over large areas of the North Slope of Alaska. It looks to be a significant advance in addressing this important issue.

We thank referee #1 for the helpful comments, hints and suggestions how to improve

our manuscript. We added these suggestions in the revised version of our manuscript. The details are documented below. We provide a supplement pdf-file, where we marked the respective changes (please see the link at the end).

**Assets**

1. Airborne campaign was generally well designed including straight tracks with multiple passes over some tracks and frequent profile ascents to find the mixed-layer structure and depth. Instrumentation and sample rate (100/s at about 70 m/s true airspeed) are appropriate to the mission. One apparent deficiency was in radiation. Although the investigators measured shortwave insolation, they did not mention measuring reflected shortwave (they did determine the albedo), or upwelling and downwelling long-wave (infrared) radiation.

The 80% upwind footprint extent of our airborne flux measurements can vary from 800 m to several kilometers. In contrast, the 80% footprint of upwelling radiation measurements corresponds to a circle of ~350 m diameter with its center located approximately below the aircraft. In result, the flux and radiation observations represent entirely different surfaces, and cannot be directly used together. For this reason we decided to use the remote sensing data, which cover the actual flux footprint. It should be noted that the sole purpose of the remote sensing data is to explain spatial variation in the environmental response function. In contrast, temporal variation is explained through downward shortwave solar radiation and observation time. The albedo was determined from the MODIS data post-processed data by the National Research Council.

2. Pg 5 line 17: "Wavelet cross-scalograms were integrated in frequency over transport scales up to 20 km" This raises the question of the length of the fight tracks. From
the map (Figure 1) they appear to be mostly about 100 km, which still provides an adequate 80 km of flux uncontaminated by edge effects at the largest width (20km?) of the cone of influence.

The transport scale of 20 kilometer for Wavelet integration was chosen based on spectral gap analysis, and exceeds the maximum observed boundary layer height by at least one order of magnitude. The eddy motions responsible for the net vertical transport are approximately confined to the scale of the boundary layer height. In result, further extending the transport scale beyond the spectral gap would effectively include compensatory flux contributions resulting from horizontal gradients, rather than actual vertical exchange. In theory, also these ultra-long scale contributions approach their ensemble value when averaged over space and time per ergodic hypothesis (Finnigan et al., 2003). However, they manifest themselves as erratic contributions to the individual flux observations, which cannot be related to surface properties in the turbulent flux footprint and thus complicate the extraction/projection of reliable relationships (Metzger, 2018).

3. Section 2.3: It's good to see a mesoscale model used in the study. In principle it can describe the mesoscale environment of the airborne campaign. It is difficult to relate model results quantitatively to what the aircraft and any fixed surface sites are reporting, especially in Alaska where the input data are relatively sparse. But for a "projection" (see item 6) it appears to work well.

Yes, the WRF model was very useful tool to improve our method to project turbulent fluxes.

4. Page 6, line 6: The verb "project" is a great word to describe the inference of

air-surface exchange over large areas from measurements over small areas because it implies some sort of model which represents knowledge and draws on data both of which grow more sparse as the scale increases. "Upscaling" is commonly used but is troublesome because it implies that the system is (largely) scale-independent, like going from a model to a full-scale prototype.

It is a very good comment, this verb directly describes what we can do with ERFs and machine learning.

5. Page 8: MODIS has something like 500 m to 1000 m pixels, which appears a good match to the km-scale spatial averages of the fluxes derived from this campaign's measurements.

Yes, it is true. Moreover, for EVI we used composite products with 250 m pixels. They are based on the best available pixel value from all acquisitions from the observed period with low clouds and low view angle.

6. Figures 5 and 6, pages 9 and 10: The "rug" plots showing the distribution of the variables in deciles are appreciated.

We found "rug" plots very useful for the interpretation of response functions.

**Questions and Issues**

1. Page 5 and thereabouts: Recognizing flight safety as the ultimate first priority, did they next prioritize near-constant height above ground or did they try to minimize

pilot adjustments? These are often a trade-off because terrain is rarely really flat. Minimizing control pressures improves fidelity of the flux measurements by minimizing the flow distortion, but usually requires higher-altitude flight to clear the terrain safely. Higher altitude gives up the advantage of cleaner sampling of heterogeneous surface fluxes.

Actually, neither of both was really enforced. Occasionally pilot action was strong, but for most of the survey flights the measuring height was nearly constant. Additionally, the Alaskan North Slope is really flat. The median terrain height and its median absolute deviation along the flight lines was 21 m±13 m and allowed us to measure at the median height 38 m with median absolute deviation ±7 m. We added this information to the manuscript (s. page 5).

2. Page 5, line 25 into Page 6: I presume that a sample of sensible or latent heat flux is one of the 1000 m mean fluxes computed every 100 m, perhaps further segregated by land-surface type. Because of the deep overlap, these are strongly autocorrelated implying fewer degrees of freedom (DF) than there are fluxes in the sample. There are ways to estimate this reduction in DF by determining the decorrelation length. A coarse guess would be to divide the sample size by 10 since the reporting interval (100 m) is one tenth of the averaging length (1 km). The loss is probably not that much. At the least the loss of DF by autocorrelation should be mentioned qualitatively.

We thank the referee for this valuable suggestion and mentioned the loss of DF in the manuscript (s. page 6).

3. Page 8, Fig 4: in the second-row left node the dashed arrow intrudes into the ellipse partially obscuring the criterion for insolation making it hard to read ($S > 380$ W/m$^2$?).

Corrected.

4. Pg 11 Figure 8 Caption: Is it proper to call the quality parameter a standard error, or is it more an uncertainty? If an error, what constitutes "truth"?

We do not mean the quality parameter of flux measurements. This figure shows the median flux maps and the "standard error" is a statistical parameter of this average equal to median absolute deviation divided by the square root from the size of the sample. We replaced the words "standard error" by "standard error of the median value".

5. Figure 7 is interesting. There are two slopes of BRT predicted vs aircraft measured. The majority of the aircraft data match the BRT predicted quite well, but the majority of the spread of BRT vs measured has a shallower slope. Away from the training data, the BRT underpredicts the strong fluxes and overpredicts the weak fluxes. Were measures taken to avoid overfitting?

At this time we did not use any methods to avoid under- or overprediction of BRT. We will take this suggestion into account for future investigations. However, we have to mention that only few data are located in the range of overfitting. For the sensible heat flux only 10% of the data are less than -5 and more than 80 W/m$^2$ and located outside of the black cloud. For the latent heat flux only 6% are less than 0 and more than 110 W/m$^2$. We added this remark to the manuscript (s. page 9-10).

6. Page 17, line 7ff: It is indeed likely that under free-convection conditions (light winds

and strong insolation) that fixed sites do not experience the larger patterns that aircraft can readily penetrate. If these structures are in fact turbulence (hence "random") they can be treated by integral techniques such as time/frequency-domain methods or time/space-domain averages. If they are not random it is not clear that an individual aircraft track in space and time will capture the relevant (nonrandom) structure unless it can be designed to do so.

Yes, they are non-random. For example, surface heterogeneity and as you mention light wind and strong isolation could result in such non-random structures, which can be captured by randomness of flight tracks. We learned it also from our machine learning method: the more data used for training, the better the convergence of the algorithm. Therefore, we always tried to extend the coverage of our airborne measurements during the reported campaign and campaigns performed in following years.

7. Page 19, lines 2 and 3: The data set is no longer unique. Sayres et al. (2017), cited in this manuscript, also used an aircraft to measure fluxes (primarily latent heat and methane) over heterogeneous Arctic landscapes. Exclusive use of airborne data can be considered a liability because of the inherently greater uncertainty in airborne measurements. Having a surface reference, as in this manuscript with Atqasuk and Happy Valley, is important both for temporal continuity and for sanity check. Data are, of course, the more readily acquired from aircraft than from fixed sites over remote areas difficult of access as on Alaska's North Slope.

We agree with this comment, but we would like to point out that this data set is unique in its spatial extent and clarified the related sentence (s. page 19).

**Copy-editing items** (a few were found and noted, not guaranteed to be exhaustive list)

1. Page 4 line 16: "reference period"

Corrected.

2. Page 14 line 15: Likely: "periods that were"

Corrected.

3. Page 14 line 20: Sentence starts with Greek letter (beta for Bowen Ratio). Recasting of the sentence is recommended.

Corrected. The sentences before and after were also corrected for the same reason.

4. Page 15 line 19 Remove "An"; insert "an": "measure water vapor <an> LI-7200 gas analyzer..."; remove "for water vapor measurements.

Corrected.

5. Page 17 line 25: Turbulent fluxes over water surfaces are more likely to be suppressed (due to lack of both mechanical and buoyant generation) than to be directed to the surface.

Yes, it is correct. We added this remark to the discussion. However, we also observed

negative sensible heat fluxes during similar campaigns in the Mackenzie River Delta, Canada, and they are also reported by Overland et al. (2000), who observed flux compensation during the SHEBA experiment.

6. Page 17 line 30: "Project" (verb) would be a really great word here instead of "upscale."

Corrected.

**Referencies:**

Finnigan, J. J., Clement, R., Malhi, Y., Leuning, R., and Cleugh, H. A.: A re-evaluation of long-term flux measurement techniques. Part 1: Averaging and coordinate rotation, Boundary Layer Meteorol., 107, 1-48, doi:10.1023/A:1021554900225, 2003.

Metzger, S.: Surface-atmosphere exchange in a box: Making the control volume a suitable representation for in-situ observations, Agric. For. Meteorol., 255, 68-80, 2018.

Overland, J. E., McNutt, S. L., Groves, J., Salo, S., Andreas, E. L., and Persso, P. O. G.: Regional sensible and radiative heat flux estimates for the winter Arctic during the Surface Heat Budget of the Arctic Ocean (SHEBA) experiment, J. Geophys. Res., 105, 14 093 – 14 102, 2000.

Please also note the supplement to this comment:

https://www.atmos-chem-phys-discuss.net/acp-2017-1166/acp-2017-1166-AC1-supplement.pdf
* * *

---

## Author Comment (AC2) · 14 Jun 2018

**General report on Edited version**

This is an excellent combination of aircraft flux measurements, machine learning algorithms extrapolated to surface response functions. It also links to large scale modeling to allow contextual interpretation with respect to hydrological and energy budget response focused on an important climate sensitive region. It provides a
significant advance in the area of airborne flux measurements and relevance to validation of surface response function dependent models. This is an important paper as this very thorough approach to airborne eddy covariance fluxes has really been missing from the scientific literature over the past 1-2 decades in general. The regional model comparison is a nice addition to capture the mesoscale variability and scales although the study is limited by the surface data availability. I do like the terminology used for model-aircraft comparison.

We would like to sincerely thank the referee #2 for the evaluation and the constructive comments on the manuscript. Our responses to the comments and explanations how we revised the manuscript are documented below. We provide a supplement pdf-file, where we marked the respective changes (please see the link at the end).

**Updates**

Relevant discussion on the design and implementation of the aircraft campaign is included and is sufficient for replication and addressing of issues and potential artifacts in such approaches.

The methodologies are very well described and relevant to the technique applied. These are appropriate to the conclusions arrived at with some limitations however in completing the full energy budget. These could have been discussed further with respect to the uncertainties but generally I don't think this could be improved on.

As you already mention, this paper is our first attempt to apply the methods like airborne eddy-covariance (EC) measurements, mesoscale modeling and machine learning for the projection of energy fluxes. An inclusion of additional data for the

entire North Slope with high resolution in space and time requires a lot of efforts in computing time and handling of the big data. However, lessons learned allow us to do this for future studies and we are going to include more parameters from the model like other radiation components, ground fluxes, residual of energy budget.

The relevant transform scales based on the flight track described appear consistent with the approach and is explained well and are also consistent with results previously published in the literature (although these were limited in terms of surface site comparisons). The relevant edge effects associated with the wavelet analyses are always an issue but I think these are within the uncertainties when scaling to the regional observations and looks quite reasonable. Whilst data quality control is critical for such wavelet analyses and could be quantified further I think we can assume this is good based on the results. There are other transform approaches that could have been compared but likely these would not have changed the results.

The transform scales were chosen based on flight lengths as well as on spectral gap analysis (please see our response to reviewer #1, asset 2). Sayres et al. (2017) and Dobosy et al. (2017) recently published comprehensive comparisons of transforms for estimation of turbulent fluxes using airborne measurements. The other transforms may improve the quality of flux measurements. But we were mostly focused on the flux projection on a regional scale and guess, that the most significant improvement of the result uncertainty will be achieved by addition of new flights covering different areas of the North Slope and different meteorological situations.

Page 8: The addition of the distribution "rug" plot, Figure 5, is very useful.

We found "rug" plots also very useful for the interpretation of response functions.

**Minor Questions and Formatting Issues**

Item 1. Some brief comment on the appropriate optimization of relevant straight and level sampling altitudes for the flux measurements (discussed page 5 etc) with respect to heterogeneity scales within the flight track would be helpful but not essential here?

We added a paragraph about the heterogeneity of the Alaskan North Slope and appropriate pilot action (see page 5).

Item 2. Figure 4. Some of the arrows in the boosted regression tree figure overlap/obscure the text in the various nodes, e.g. $a>0.5$, $S\downarrow>380$ W/m$^2$, $r>7$ k kg$^{-1}$

Corrected.

Item 3. Figure 7. It appears obvious that there are two clusters within the sensible and latent heat flux (predicted versus aircraft measured) domain with significantly different slopes with under-predictions at high values in each case. Can the authors comment on this? Is there a potential bias here?

The under- and overestimation is also mentioned by the referee #1 (please see "Questions and Issues, #5"). The clouds of overfitting appear due to an insufficiency of measurements with high and low energy fluxes for machine learning in this range. Most of the data are located in the black cloud. For the sensible heat flux only 10% of the data are less than -5 and more than 80 W/m$^2$ and located outside of the black cloud. For the latent heat flux only 6% are less than 0 and more than 110 W/m$^2$. We

added this remark to the manuscript (see page 9-10).

Item 4. Figure 8. Legend: Has the standard error used in this figure been defined?

We do not refer in the legend to any quality parameter of flux measurements. Figure 8 shows the median maps and the "standard error" is a statistical parameter of this average equal to the median absolute deviation divided by the square root from the size of the sample. We replaced the words "standard error" by "standard error of the median value" to avoid misunderstanding.

Item 5. As mentioned, the impact of enhanced convective conditions suggests potential under-sampling bias of all relevant scales in these conditions. It would be useful to mention the range therefore where such comparisons may break down, but this may require more detailed spectral analysis for another discussion. However I think this caveat/statement addresses the issue adequately for the work presented.

Please see our response to Item 3.

Item 6. Figure 7 is a brave plot (and we need more of them in the literature before relying overly on tower data). I think the discussion and literature references regarding the discrepancies with WRF are adequate but do highlight that there is still a great deal of work to do here.

Please also see our response to Item 3. We agree and see a lot of opportunities to improve our knowledge about surface turbulent exchange combining process-based mesoscale models, projection of regional airborne measurements and small scale EC

tower data.

Final comment: The authors are to be commended for delivering an excellent set of results.

**Referencies:**

Dobosy, R., Sayres, D., Healy, C., Dumas, E., Heuer, M., Kochendorfer, J., Baker, B., and Anderson, J.: Estimating random uncertainty in airborne flux measurements over Alaskan tundra: Update on the Flux Fragment Method, J. Atmos. Oceanic Tech., 34, 1807–1822, https://doi.org/10.1175/JTECH-D-16-0187.1, 2017.

Sayres, D., Dobosy, R., Healy, C., Dumas, E., Kochendorfer, J., Munster, J., Wilkerson, J., Baker, B., and Anderson, J.: Arctic regional methane fluxes by ecotope as derived using eddy covariance from a low-flying aircraft, Atmos. Chem. Phys., 17, 8619–8633, https://doi.org/10.5194/acp-17-8619-2017, 2017.

Please also note the supplement to this comment:
https://www.atmos-chem-phys-discuss.net/acp-2017-1166/acp-2017-1166-AC2-supplement.pdf

**Supplement:**

[revised manuscript text omitted]